# Decomposition of Flavonols in the Presence of Saliva

**Malgorzata Rogozinska and Magdalena Biesaga ***

Faculty of Chemistry, University of Warsaw, Pasteura 1, 02-093 Warsaw, Poland; mgwiazdon@chem.uw.edu.pl
* Correspondence: mbiesaga@chem.uw.edu.pl

**Abstract:** In this study, the LC-MS/MS was applied to explore the stability of four common dietary flavonols, kaempferol, quercetin, isorhamnetin, and myricetin, in the presence of hydrogen peroxide and saliva. In addition, the influence of saliva on the representative quercetin glycosides, rutin, quercitrin, hyperoside, and spiraeoside was examined. Our study showed that, regardless of the oxidative agent used, flavonols stability decreases with increasing B-ring substitution. The decomposition of analyzed compounds was based on their splitting by the opening the heterocyclic C-ring and realizing more simple aromatic compounds. The dead-end products corresponded to different benzoic acid derivatives derived from B-ring. Kaempferol, quercetin, isorhamnetin, and myricetin were transformed into 4-hydroxybeznoic acid, protocatechuic acid, vanillic acid, and gallic acid, respectively. Additionally, for quercetin and myricetin, two intermediate depsides and 2,4,6-trihydroxybenzoic acid derived from A-ring were detected. All analyzed glycosides were resistant to hydrolysis in the presence of saliva. Based on our data, saliva was proven to be a next oxidative agent which leads to the formation of corresponding phenolic acids. Hence, studies on flavonols' metabolism should take into consideration that the flavonols decomposition starts in the oral cavity; hence, in subsequent parts of the human digestive tract, they could be present not in their parent form but as phenolic acids. Further analyses of the influence of saliva on flavonols glycosides need to be performed due to the possible interindividual fluctuations.

**Keywords:** flavonols; $H_2O_2$; saliva; metabolism; oxidation; LC-MS/MS

## 1. Introduction

Flavonoids are phenolic compounds widely present in plants and food of plant origin. Both clinical and epidemiological studies show the correlation between the dietary polyphenols intake and the reduction of risk of some chronic diseases such as cardiovascular diseases, cancer, and diabetes, as well as aging [1–4]. These beneficial effects are associated with their antioxidant activity [5–7]. Flavonols, the major class of flavonoids present in the human diet, and among them quercetin, kaempferol, myricetin, and isorhamnetin (Figure 1), are well known to act as antioxidants in vitro and show protective effects against free radicals, reactive oxygen species, and other oxidation agents [8–10]. However, the biological properties of antioxidants such as flavonols depend on their bioavailability and metabolism in the human body. The study of biological responses due to dietary intake of polyphenols cannot be carried out without taking into consideration polyphenols–saliva interactions. Although phenolics are normally ingested through the mouth as elements of food, very little is known about their metabolism in the oral cavity. One of the main goals assigned to saliva is participation in glycosides hydrolysis [11,12], which delivers biologically active aglycones that can be absorbed more effectively in the human digestive tract. Since flavonols mainly occur in food as glycosides, much of the research focused on the metabolism in oral cavity concerns glycosides. However, some reports show the presence of flavonols aglycones in the food of plant origin, for instance in eucalyptus and unifloral types of honey [13]. Moreover, the fermentation carried out during the manufacturing of food could

results in the hydrolysis of glycosides to aglycones. Although it was reported that flavonoid glycosides can be absorbed intact via the sodium-dependent glucose transporter [14], it was also shown that many glycosides are not absorbed due to efficient efflux transport by intestinal efflux protein pumps [15].

| Flavonol | $R_1$ | $R_2$ |
|---|---|---|
| **Kaempferol** | H | H |
| **Quercetin** | OH | H |
| **Isorhamnetin** | $OCH_3$ | H |
| **Myricetin** | OH | OH |

**Figure 1.** Structures of kaempferol, quercetin, isorhamnetin, and myricetin.

It is well established that the oral cavity harbors numerous and diverse microorganisms, which can hydrolyze flavonols glucosides to the aglycones by glucosidases excreted from the bacteria [11,16]. All of these features indicate that the effective absorption of polyphenols in the human digestive tract strongly depends on their deglycosylation, and, as far as we know, study on the metabolism of aglycones in the oral cavity are limited. Interaction of saliva components with bioactive compounds from food occurs due to various reasons. Firstly, chewing food allows its components to stay in the oral cavity for a while, which ensures the contact of saliva and oral mucosa with food. Secondly, during chewing, flavonols are dissolved in saliva, which facilitates their interaction with oral mucosa and salivary proteins [17–20]. It is worth noting that Quijada-Morín et al. [17] outlined interactions between flavan-3-ols and salivary proteins not only as a precipitation issue as it has been usually studied but also as a more complex interaction, which involves the formation of soluble and insoluble complexes.

All of these mechanisms increase lipophilic polyphenols assimilation and causes their retention in the oral cavity over time [21–23].

The human oral cavity contains numerous and diverse microorganisms as commensals [24,25]. Approximately 280 bacterial species from the oral cavity have been isolated in culture and formally named. The possible reason for the decomposition of flavonols in the presence of saliva is the fact that bacteria and leucocytes presented in the oral cavity are able to generate $H_2O_2$; thus, flavonols could be easily oxidized [26,27]. While there is a lack of reports showing the oxidation of flavonols in the presence of saliva, other oxidizing conditions have been well established, mainly for quercetin. Quercetin is degraded in air conditions with the formation of different depsides, as intermediates in the degradation pathway. Further decomposition results in the formation of 2-(3,4-dihydroxyphenyl)-2-oxoacetic acid, 2,4,6-trihydroxybenzoic acid, and 3,4-dihydroxybenzic acid [28]. Additionally, Maini et al. [29] proposed another degradation products of quercetin after its exposure to UVA radiation: 1,3,5-trihydroxybenzene and 2,4,6-trihydroxybenzaldehyd. In general, oxidation of quercetin by various methods: air, electrochemical, enzymatic, and free radical oxidation may yield, more or less, the same set of oxidized products [30].

For further elucidation of the oxidation processes of flavonols in biological systems, we investigated the stability of four wide-spread flavonols: quercetin, kaempferol, myricetin, and isorhamnetin in the presence of saliva. This work also provides comparative oxidative studies of flavonols using $H_2O_2$ solution and whole saliva as oxidation agents. We chose flavonols that differ in the number of hydroxyl substituents in the B-ring. Additionally, in the case of isorhamnetin, one hydroxyl group is replaced with a methoxy group, which allows for the blocking of interactions from two adjacent hydroxyl groups. To determine the various degradation products of flavonols, LC-MS/MS system was applied.

## 2. Materials and Methods

### 2.1. Chemicals and Reagents

The commercial standards of flavonols and phenolic acids, as well as the rest of the chemicals, were purchased from Sigma-Aldrich (Steinheim, Germany). Glycosides rutin (quercetin 3-rhamnoglucoside), quercitrin (quercetin 3-rhamnoside), hyperoside (quercetin 3-galactoside), and spiraeoside (quercetin 4′-glucoside) were purchased from Extrasynthese (Genay, France). Methanol was obtained from Merck (Darmstadt, Germany). In all experiments, ultrapure water from a Milli-Q system with an electrical resistivity of 18 MΩ × cm was used. Stock solutions of flavonoids were prepared in methanol. Diluted standards were prepared in 25% methanol with a final concentration of 50 mg/L. All solutions were filtered through 0.45 μm membranes (Millipore) and degassed prior to use.

### 2.2. Collection of Saliva Samples

Five healthy human volunteers (20–30 years old) were recruited to this study based on restricted criteria. All volunteers were self-reported to be in good general health without any chronic diseases, not taking antibiotics, and no history of drinking and smoking habits. Volunteers were requested to refrain from eating for at least 10 h and they could drink only mineral water before collection. Saliva samples were collected in the morning 2 h after brushing teeth with toothpaste free of polyphenols. Before saliva collection, the oral cavity was rinsed with ultrapure water to remove dead skin cells. Saliva was mechanically stimulated by chewing a plastic tube and collected into the plastic containers. Then, individual saliva samples (n = 5) were diluted with water (1:1, w:w) and incubated at 37 °C in a water bath until analysis.

### 2.3. Influence of Saliva/$H_2O_2$ Solution on Stability of Flavonols

Standards of kaempferol, isorhamnetin, quercetin, and myricetin were mixed with prepared saliva or $H_2O_2$ solution at 25 mg/L in 12.5% of MeOH final concentration and incubated for 24 h at 37 °C. Samples were prepared to obtain the ratio 1:1 v:v of saliva or equimolar $H_2O_2$ to the standard solution. The ratio of saliva and standard solution was evaluated to obtain optimal flavonols decomposition time, which allowed detecting unstable intermediates [31]. Since the non-glycosylated flavonols did not easily dissolve in the water, addition of MeOH had to be used, and the minimum concentration of MeOH was 25% in stock solution (12.5% in sample solution). Additionally, each flavonol was mixed with the same volume of the water, as a control sample. All samples were filtered through 0.45-μm PTFE membranes and analyzed at different times of incubation over 24 h. A long incubation period allowed us to find the dead-end products of flavonols degradation. The pH value of all solutions was controlled and measured at the beginning and end of the reaction.

### 2.4. Influence of Microbiota

To check if the decomposition of flavonols is the result of the presence of microbiota, we decided to incubate quercetin (as a representative flavonol) with saliva filtered using sterile polyethersulfone (PES) membrane (pore size 0.22 μm) to remove bacteria from the sample. For that purpose, 5 mL of saliva was diluted 1:1 with distilled water and shaken vigorously to reduce viscosity. In parallel, 100 μL of filtered saliva was spread onto 5% blood agar and incubated for 24 h at 37 °C in duplicate to examine if the usage of 0.22 μm membranes ensures the sterility of the sample. At the same time, the experiment with the incubation of quercetin with bot saliva filtered using 0.22 and 0.45 μm membranes was carried out.

## 2.5. Apparatus

The analytical method used in the presented work was developed in our laboratory and discussed in detail in a previous paper on the study of the phenolic compounds [32]. Chromatographic analyses were carried out using LC-MS/MS system consisted of binary pumps LC20-AD, degasser DGU-20A5, column oven CTO-20AC, and autosampler SIL-20AC, connected to 3200 QTRAP Mass spectrometer (Applied Biosystem/MDS SCIEX). Compounds were separated on Chromolith Performance C18 column (100 × 2mm, Merck) at 30 °C. Formic acid (8 mM, pH 2.8) as eluent A and methanol as eluent B were used. Samples incubated at 37 °C were kept in the same temperature before analysis in an autosampler. The flow rate of mobile phase was 0.2 mL/min and the gradient mode was as follow: 0–3 min, 10% B; 20–25 min, 50% B; 26–40 min, 10% B. LC system was connected to the 3200 QTRAP Mass spectrometer (Applied Biosystem/MDS SCIEX) with electrospray ionization (ESI) working in negative mode. ESI conditions were as follows: capillary temperature of 450 °C, curtain gas at 0.3 MPa, auxiliary gas at 0.3 MPa, and negative ionization mode source voltage of 4.5 kV. Nitrogen was used as curtain and auxiliary gas. Analyst 1.4.2 software was used for data acquisition. LC-MS/MS analysis were carried out by comparing retention time and *m/z* values obtained by MS and MS$^2$ with the mass spectra of standards obtained under the same conditions. Because some degradation products such as depsides are not commercially available, the presence of these compounds was confirmed by comparison of retention times, masses, and fragmentation spectra of potential oxidation products with literature.

Quantification of compounds was done using the selected reaction monitoring mode (SRM). For each compound, the optimum conditions for SRM mode were determined in infusion mode and two SRM pairs were chosen as representatives (SRM1 and SRM2) (Table 1). Due to the higher intensity of peak obtained using the SRM1 pairs, they were chosen for qualitative analyses. Calibration curves were drawn from the analysis of 5 μL volumes at concentration ranging from 0.5 to 50 mg/L (n = 7) measured in triplicated. Coefficients of linearity (R$^2$) for the calibration curves were ≥ 0.996. LODs were estimated by decreasing the concentration of the analytes to the smallest detectable peaks, and then its concentration was multiplied by three. LODs ranged 0.1–0.5 mg/L.

**Table 1.** LC-MS/MS characteristics of phenolic compounds in the negative mode.

| Compound | Retention Time, Min | Q1 Mass | Q3 Mass | DP, V | CE, V |
|---|---|---|---|---|---|
| Gallic acid | 3.0 | 169 | 125 | −45 | −20 |
| | | | 97 | −45 | −26 |
| Protocatechuic acid | 4.4 | 153 | 109 | −15 | −18 |
| | | | 66 | −15 | −26 |
| 4-hydroxybenzoic acid | 7.1 | 137 | 93 | −25 | −18 |
| | | | 119 | −25 | −8 |
| Vanillic acid | 9.8 | 167 | 152 | −30 | −16 |
| | | | 107 | −30 | −24 |
| Hyperoside | 16.6 | 463 | 300 | −80 | −32 |
| | | | 271 | −80 | −56 |
| Spiraeoside | 18.4 | 463 | 300 | −80 | −32 |
| | | | 151 | −80 | −48 |
| Rutin | 18.9 | 609 | 300 | −65 | −32 |
| | | | 270 | −65 | −74 |
| Myricetin | 20.1 | 317 | 151 | −20 | −26 |
| | | | 179 | −20 | −22 |
| Quercitrin | 20.4 | 447 | 300 | −60 | −26 |
| | | | 270 | −60 | −56 |
| Quercetin | 22.8 | 301 | 151 | −40 | −30 |
| | | | 179 | −40 | −24 |
| Kaempferol | 25.0 | 285 | 151 | −45 | −25 |
| | | | 117 | −45 | −53 |
| Isorhamnetin | 25.6 | 315 | 300 | −35 | −24 |
| | | | 199 | −35 | −32 |

*2.6. Statistical Analysis*

The statistical analyses of the data were carried out using Microsoft Excel 2016 and Excel's Analysis Toolpak (ANOVA). The one-way analysis of variance (ANOVA) and the significance of differences between sample means were calculated, and $p$ values $\leq 0.05$ were taken into account as significant.

## 3. Results and Discussion

*3.1. Oxidation of Flavonols with Hydrogen Peroxide*

As compared to the control, a noticeable decrease of quercetin concertation was observed during its incubation in the presence of $H_2O_2$. As shown in Figure 2, rapid degradation of quercetin could be observed with the simultaneous formation of three other compounds with $[M-H]^-$ at $m/z$317 ($t_r = 17.5$ min) $[M-H]^-$ at $m/z =193$ ($t_r = 4.4$ min) and $[M-H]^-$ at $m/z$ 169 ($t_r = 5.0$ min). Oxidation product with $[M-H]^-$ at $m/z$ 317, identified as depside characteristic for quercetin, was found as an intermediate product, the concentration of which increased in the first 80 min of the experiment. Further incubation led to the decrease of its concentration, whereas new peaks with $[M-H]^-$ at $m/z$ 193 ($t_r = 4.4$) and $[M-H]^-$ at $m/z$ 169 ($t_r = 5.0$) appeared. Final degradation products were identified as 2,4,6-trihydroxybenzoic acid ($m/z$ 169) and protocatechuic acid ($m/z$ 193). Such results suggest that the oxidation of quercetin with $H_2O_2$ is based on its splitting by the opening the heterocyclic C-ring and realizing simpler aromatic compounds. Moreover, oxidation involves the initial oxidative step with subsequent changes in the flavonol skeleton such as the formation of B-ring orthoquinone and rearrangement in the C-ring [33]. Detected oxidation products are similar to those obtained under other conditions such as oxygen [28], UVA and UVB [34], hydroxyl free radical [35], and presence of copper (II) [36,37]. Thus, the hypothesis of Zhou et al. [30] that the oxidation of quercetin using different oxidations agents may yield, more or less, the same set of oxidized products seems to be particularly relevant. Decomposition pattern of myricetin leads throughout the characteristic depside as an intermediate at $m/z$ 321. The LC-MS/MS measurements show that further oxidation led to its decomposition and formation of gallic acid as a corresponding hydroxybenzoic acid derivative at $m/z$ 169 and $t_r = 2.9$ min. Besides, 2,4,6-trihydroxybenzoic acid ($t_r = 4.9$ min) was detected as a degradation product of myricetin. Although it shares the same mass as gallic acid, their separation was obtained in the established method. Unfortunately, neither for isorhamnetin nor for kaempferol corresponding depsides were detected. Nevertheless, transformations which involve initial oxidative steps with subsequent changes in the flavonols skeleton was observed. As a result of kaempferol oxidation, the formation of a compound with $t_r = 7.1$ min and SRM characteristic for 4-hydroxybenzoic acid (137/93) was observed. The breakdown product of isorhamnetin was identified as a vanillic acid with retention time 9.8 min and SRM pair (167/152). These results suggest that, as in the case of quercetin and myricetin, oxidation of kaempferol and isorhamnetin leads to C-ring cleavage in the flavonols' structure. As a result of this reaction, the corresponding hydroxybenzoic acid derivatives are formed. It should be mentioned that two reactive centers in the C-ring were identified as responsible for antioxidant activity in flavonols: the 2,3 double bond in conjugation with the 4-oxo function and the 3- and 5-hydroxyl groups with hydrogen bonding to the same 4-oxo function (for flavonols numbering, see Figure 1) [38,39]. That is why all of the modifications which occur in that area could significantly alter the chemical and biological properties of flavonols.

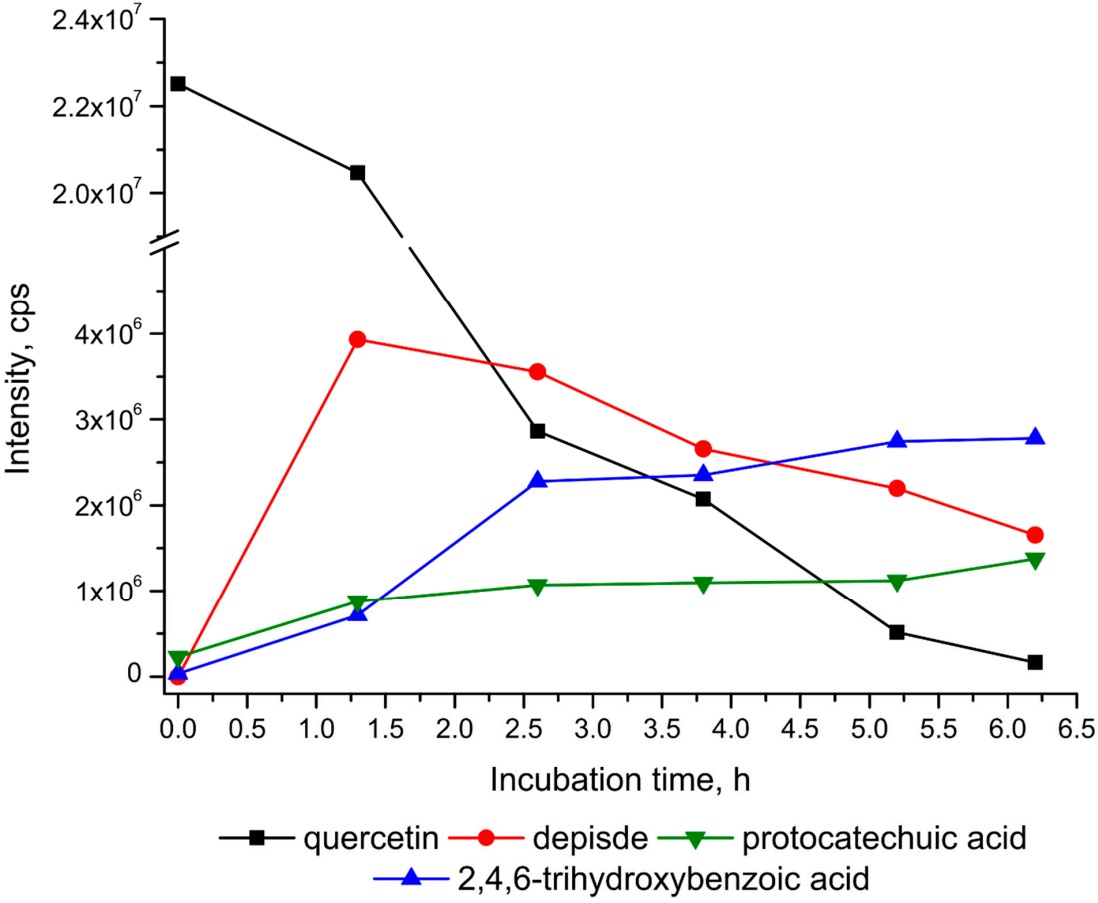

**Figure 2.** Degradation profile of quercetin and its main products formed during the incubation with $H_2O_2$ solution.

### 3.2. Stability of Flavonols in Saliva Solution

Figure 3 presents the loss of the starting amount as percent of remaining kaempferol, quercetin, and isorhamnetin (Figure 3a) as well as myricetin (Figure 3b) for a single representative after 6 h of incubation with saliva. The levels of the examined compounds were significantly different after 6 h of incubation. Kaempferol was the most stable compound in saliva during this time (about 73% left). Quercetin and isorhamnetin were less stable than kaempferol, and the degree of their degradation was 52% and 59%, respectively. Myricetin was the least stable flavonol under these conditions, and its concentration was at trace levels at the end of the experiment. As can be readily seen, myricetin showed rapid degradation and disappeared after 90 min of incubation. The general order of stability of examined flavonols was as follows for all saliva samples collected from five human volunteers: kaempferol > isorhamnetin > quercetin >> myricetin. Generally, the decomposition rate of flavonols increases with an increasing number of hydroxyl groups attached to the B-ring. A similar order of degradation of these flavonols was observed in $H_2O_2$ solution. These results are in good agreement with observations of Maini et al. for ultraviolet radiation A (UVA) [29]. During the incubation of quercetin with saliva solution, the decomposition of quercetin into compounds with *m/z* 305 and *m/z* = 317 was observed. These two peaks were identified as the intermediate products of quercetin oxidation (depsides) and their formation under oxidative conditions has already been described [28,29,33]. During further incubation, depsides decomposed with a simultaneous appearance of another peak with the retention time 4.41 min, mass spectrum, and SRM pair (153/109) characteristic for protocatechuic acid (Figure 4). In contrast to the incubation of quercetin in the presence of $H_2O_2$ mixture, the formation of 2,4,6-trihydroxybenzoic acid as a degradation product was not observed.

Similar results were obtained for myricetin. However, the latter was less stable in the presence of saliva than quercetin. Chromatogram of myricetin incubated with saliva showed a decrease in its peak's intensity, whereas a new peak at *m/z* = 321 appeared. This peak was considered as depside characteristic for myricetin. Moreover, the formation of a new compound with retention time (t$_r$ = 2.9 min), mass spectrum, and SRM pair characteristic for gallic acid (169/125) was observed. Unfortunately, as mentioned above, among studied flavonols, only for myricetin and quercetin the corresponding depsides were detected. Chromatograms and mass spectrum obtained for isorhamnetin and kaempferol showed their degradation to vanillic and 4-hydroxybenzoic acids, respectively. In an oxidation experiment, Maini et al. [29] suggested that the presence of the corresponding phenolic acid derivative in the absence of any detectable depside concentration is the result of comparable depside formation and hydrolysis rates. Krishnamachari et al. suggested that the presence of both a catechol unit in the B-ring and a free C-3 hydroxyl appears to be a prerequisite for the formation C-ring carbocation or *p*-quinone methide (which formation proceeds predominantly through its tautomer, *o*-quinone) in the oxidative decomposition of flavonols [33,40]. It has been also demonstrated using EPR spectroscopy that the spin distribution during oxidation of quercetin remains entirely on the B-ring, promoting the donation of two electrons leading to the formation of an *o*-quinone [41]. These phenomena explain our observations that myricetin and quercetin decompose rapidly and respective depsides could be observed. Moreover, hydroxyl or methoxy substituents are considered to stabilize the flavonol C-ring carbocation intermediate. As has already been proven, a relatively electron-rich derivative may be more stable and hence more easily formed than its electron-poor analog [42]. Hence, electron-withdrawing and electron-donating groups such as hydroxyl and methoxy groups attached directly to B-ring should influence the rate of *p*-quinone methide formation. In addition, two or more electron-donating groups greatly facilitate its initial generation and stability [42]. Additionally, isorhamnetin due to the presence of a methoxy group greatly enhances the electron-donating properties in the 4'-position [38]. Finally, the presence of corresponding benzoic acid derivatives, as well as 2,4,6-trihydoxybenzoic acid, highlights possible interconversion of the *p*-quinone methide into C-ring carbocation intermediate and its further decomposition. Poorly substituted flavonols such as kaempferol with one hydroxyl group in B-ring skeleton are not able to form *o*-quinone or its tautomeric form *p*-quinone methide species readily. This theory could also explain why any detectable levels of depsides were observed for kaempferol and isorhamnetin.

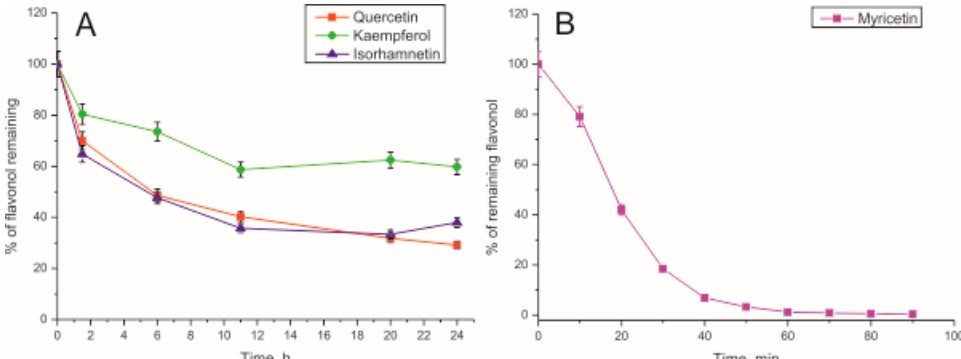

**Figure 3.** The loss of the starting amount in saliva solution as percent of remaining: (**A**) kaempferol, quercetin, and isorhamnetin; and (**B**) myricetin.

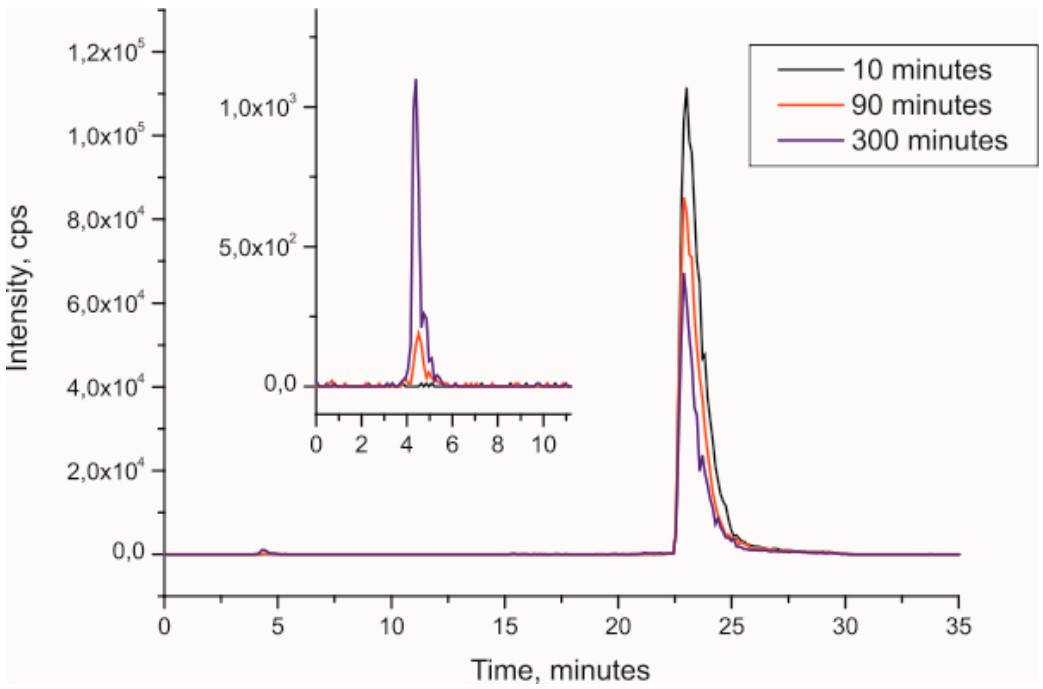

**Figure 4.** Changes in amount of quercetin (22.8 min) and protocatechuic acid (4.5 min) during incubation of quercetin with saliva.

As mentioned above, saliva was proven to contain phenolics after consumption of phenolics-rich beverages, and these compounds have been found to persist in oral cavity up to 300 min, despite a constant salivary flow [21]. Our research shows that this time is sufficient for the partial decomposition of kaempferol, quercetin, and isorhamnetin and complete decomposition of myricetin. To check if detected compounds may be considered as a dead-end product, we decided to incubate analyzed flavonols for further 18 h. After 24 h of incubation, no additional compounds were detected as degradation products of the analyzed flavonols. However, further decomposition of kaempferol, quercetin, and isorhamnetin was observed. The levels of remaining flavonols calculated as average amount for five volunteers was as follow: 55.42% ± 8.12 of kaempferol, 23.17% ± 5.48 of quercetin, and 39.37% ± 11.11 of isorhamnetin. This indicates that, even after long time of incubation, the order of stability of flavonols remains the same. Since it is known that the stability of flavonoids strongly depends on the pH, we controlled it during the experiment. For all solutions, pH fluctuation ranged from 6.96 ± 0.10 to 7.06 ± 0.15 at the beginning and from 6.98 ± 0.12 to 7.39 ± 0.09 at the end of the experimental period.

Several studies have reported that saliva can hydrolyze flavonoids glycosides, hence we checked four quercetin glycosides: rutin (quercetin 3-rhamnoglucoside), quercitrin (quercetin 3-rhamnoside), hyperoside (quercetin 3-galactoside), and spiraeoside (quercetin 4′-glucoside). Preliminary studies showed that all of the analyzed glycosides were stable in the presence of saliva. The results obtained for rutin and quercitrin are in good agreement with previous studies, which showed that these two glycosides were hydrolyzed very slowly or were resistant to salivary hydrolysis [12,31]. On the other hand, lack of spiraeoside hydrolysis was inconsistent with experiments which suggested that glucose conjugates are rapidly hydrolyzed to corresponding aglycones [11,12]. However, in the same study, a large interindividual variability in hydrolysis rate was also observed. According to the aim of our study, it is interesting to note that sugar moiety attached to flavonol inhibits C-ring decomposition.

The presented studies allow creating the general scheme of flavonols' degradation in the presence of two oxidation agents: $H_2O_2$ and saliva (Figure 5). Table 2 presents the summarized products detected in samples after oxidation with $H_2O_2$ and saliva.

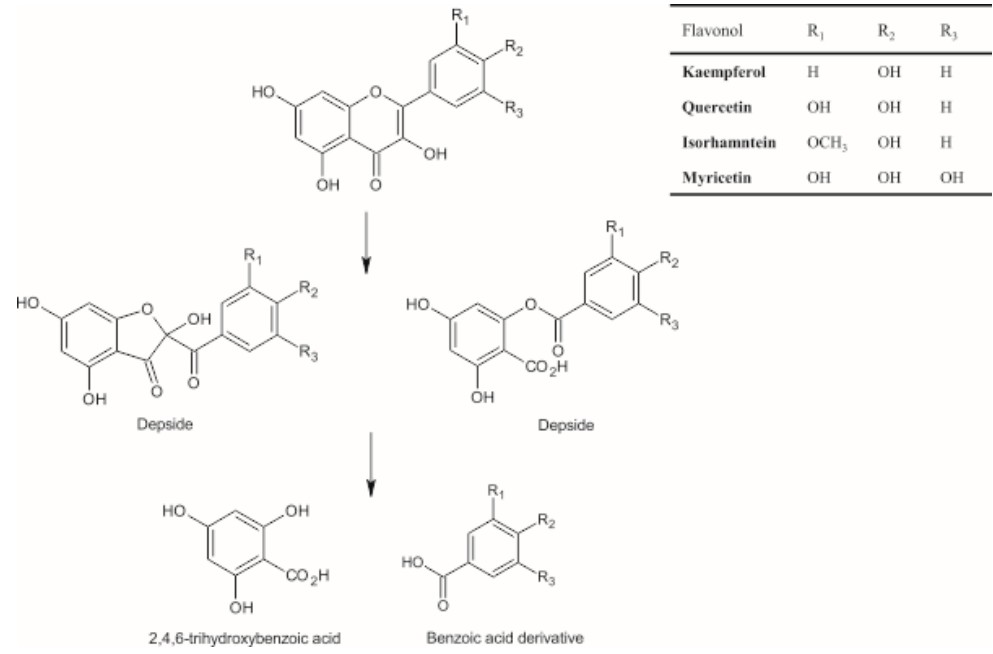

**Figure 5.** General scheme of flavonols' degradation in the presence of saliva.

**Table 2.** Comparison of products detected with $H_2O_2$ and saliva oxidation.

| | Kaempferol | | Quercetin | | Isorhamnetin | | Myricetin | |
|---|---|---|---|---|---|---|---|---|
| | $H_2O_2$ | Saliva | $H_2O_2$ | Saliva | $H_2O_2$ | Saliva | $H_2O_2$ | Saliva |
| Depside 1 | − | − | + | + | − | − | − | − |
| Depside 2 | − | − | − | + | − | − | + | + |
| 2,4,6−trihydroxybenzoic acid | − | − | + | − | − | − | + | − |
| Benzoic acid derivative | + | + | + | + | + | + | + | + |

− not detected; + detected.

Overall, decomposition of flavonols is based on the cleavage of heterocyclic C-ring, with no changes in hydroxyl and methoxy substituents in A-ring and B-ring. Decomposition of myricetin and quercetin leads to the formation of gallic and protocatechuic acid, respectively, which exhibit high redox potential due to the presence of adjacent hydroxyl groups attached to the aromatic ring [43]. Contrarily, vanillic and 4-hydroxybenzoic acids were found to be less efficient in radical neutralization reaction [43].

As mentioned above, the human oral cavity contains numerous and diverse microorganisms as commensals. It is known that microbiota can split the flavonoids by opening the heterocyclic ring and releasing simpler aromatic compounds, such as hydroxyphenylacetic acids from flavonols, which could be further metabolized to derivatives of benzoic acid [44]. Taking that into consideration, we decided to incubate quercetin, as a representative flavonol, with saliva containing microorganisms (filtration through 0.45 µm filters) and without microorganisms (filtration through 0.22 µm sterile filters). In both cases, quercetin was degraded to the same extent during its incubation to protocatechuic acid, as a dead-end product. This indicates that the decomposition of quercetin is independent of the presence of microbiota.

## 4. Conclusions

As noted in this study, the stability of flavonols in the presence of saliva solution strongly depends on the number of hydroxyl groups attached to the B-ring. Indeed, flavonol stability decreases with increasing B-ring substitution. This proves that saliva is a next oxidative agent, besides UVA and UVB radiation, air, enzymes, and free radicals, which leads to the formation of corresponding phenolic acids

as dead-end products of flavonols. The obtained results indicate that myricetin is the most effective flavonol in the process of neutralizing free radicals formed in the oral cavity, due to its easy oxidation caused by the presence of three hydroxyl groups in the B-ring of this compound. Quercetin and isorhamnetin can also be regarded as quite effective antioxidants, capable of oxidizing in the presence of saliva. Nevertheless, their degradation products such as gallic acid and protocatechuic acid still possess reducing potential and are well-known antioxidant units. Rutin (quercetin 3-rhamnoglucoside) and quercitrin (quercetin 3-rhamnoside), as well as, surprisingly, hyperoside (quercetin 3-galactoside) and spiraeoside (quercetin 4'-glucoside), were resistant to hydrolysis in the presence of saliva. Nevertheless, due to the interindividual fluctuations, further analyses should be elucidated on that issue. However, the most intriguing seems to be the relatively short residence time of most foods and their bioactive compounds in the oral cavity. Even so, it should be noticed that this time is sufficient for flavonols decomposition, which clearly shows the process of flavonols metabolic transformation starts in the oral cavity. Hence, studies on flavonols metabolism should take under consideration that, in subsequent parts of the human digestive tract, they could be present not in their parent form but as phenolic acids. Consequently, information on the bioavailability and metabolic pathway of such dietary bioactive compounds is a key part in understanding their beneficial influence on human health.

**Author Contributions:** Conceptualization, M.R. and M.B.; methodology, M.R., M.B.; validation, M.R., M.B. investigation, M.R., M.B.; writing—original draft preparation, M.R.; writing—review and editing, M.B.; supervision—M.B.; All authors have read and agreed to the published version of the manuscript.

**Funding:** This research received no external funding.

**Acknowledgments:** The presented studies were carried out with the approval of Ethics and Bioethics Committee (KEIB–5/2016, Cardinal Stefan Wyszynski University in Warsaw). LC-MS/MS measurements were performed at the Laboratory of Structural Research, Faculty of Chemistry, University of Warsaw, which was established under the European Regional Development Grant WPK_1/1.4.3./2004/72/72/165/2005/U. The authors thank prof. Dorota Korsak for her help in the microbiological part of the presented manuscript and prof. Krystyna Pyrzyńska for her helpful advice on various technical issues examined in this paper.

**Conflicts of Interest:** The authors declare no conflict of interest.

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
