# Peer review of "Decomposition of Flavonols in the Presence of Saliva"

_applsci, doi:10.3390/app10217511_

Round 1

Reviewer 1 Report

The article is aimed at the investigation of flavonols decomposition in saliva. The study is rather well designed and well described. There are some points that need to be improved before publication.

My main concern is if H2O2 concentration applied in the study reflects the concentration present in saliva. Some references showing H2O2 concentrations generated by oral bacteria should be introduced and proper discussion added.

l.17 change “date” to “data”

l.24 Introduction section is not numbered

l.102 Volunteers were not eating for 10 hours and they still had food debris in their mouths? I imagine that samples were collected in the morning and volunteers brushed their teeth. How teeth brushing influenced saliva composition or representativeness?

Author Response

Reviewer 1.

In the beginning, we would like to thank you for the insightful revision of our manuscript. We would like to respond to your comments and reviews. We detailed revisions that have been made, citing the line number and exact change. We hope that you will be satisfied with our comments and corrections.

My main concern is if H2O2 concentration applied in the study reflects the concentration present in saliva. Some references showing H2O2 concentrations generated by oral bacteria should be introduced and proper discussion added.

The H2O2 is excreted by oral bacteria and by host cells in amounts which vary with the state of cellular metabolism, the diet and other factors [1]. What is more, saliva contains different antioxidant systems includes various molecules (e.g. uric acid and ascorbic acid) and enzymes (e.g. peroxidase). In the enzymatic salivary antioxidant system,  peroxidase is by far the most significant enzyme and has a dual role: it controls the level of hydrogen peroxide excreted by bacteria (mainly by different Streptococcus) and leukocytes from the salivary glands into the oral cavity [2].  Due to this complex matrix with different redox potentials, it seems to be hard to determine the concentration of the H2O2 in saliva with classical e.g. titration or instrumental analyses.

To our knowledge, the amount of articles concerning the data about the H2O2 concentration in saliva is limited. Uehara et al. [3] determined that the mean concentration of H2O2 produced by 18 strains of viridans group streptococci (1x108 cfu in 200 µL of culture medium) was 1.24 ± 0.60 mmol. In our research, the calculated amount of bacteria corresponds to 2.2 x107 cfu/mL of saliva. However, mentioned colonies factor units cannot be compared due to the different conditions applied in experiment mainly, in our experiment we decided to harvest the bacteria culture form body fluid sample (saliva), while the authors used pure bacteria strains grown in the liquid broth. Moreover, Hamon and Klebanoff [4] noticed that using the formate oxidation method (the conversion of [14C]formate to 14CO2 by catalase and H2O2 as a measure of H2O2 formation) provides linearity response only in some ranges of the number of organisms/mL of the used broth.

As it was mentioned in our manuscript l. we decided to mix the equimolar amounts of H2O2 and flavonols in order to obtain results which could show the degradation of flavonols with the simultaneous determination of some intermediates which are not stable in oxidative conditions. The determination of the concentration of excreted H2O2 by microorganisms present in saliva seems to be really interesting and for further studies, we try to introduce this feature. However, this must be preceded by the identification of bacteria present in human saliva.

l.17 change “date” to “data”

Revised as requested

l.24 Introduction section is not numbered

The introduction section is numbered in the version found at the link sent in the e-mail.

l.102 Volunteers were not eating for 10 hours and they still had food debris in their mouths? The phrase “food debris”  was changed to “dead skin cells” l.107

I imagine that samples were collected in the morning and volunteers brushed their teeth.

Samples were collected in the morning two hours after brushing teeth with toothpaste free of polyphenols (this sentence was also mentioned in the revised manuscript l.105-106)  

How teeth brushing influenced saliva composition or representativeness?

The components of toothpaste can influence saliva composition. Thus, our procedure of saliva collection was optimized and was strictly followed by each volunteer.

Following steps were introduced to reduce the fluctuation of saliva composition:

  • the usage of the same toothpaste free of polyphenols in which the number of ingredients was reduced to a minimum,
  • collection of saliva samples after the same period (2 hours) from teeth brushing,
  • rinsing the oral cavity with ultrapure water before saliva collecting.

Due to the same procedure followed by each volunteer, it was assumed that potential changes in saliva composition could be considered as a systematic error.

  1. Tenovuo, J.; Pruitt, K.M. Relationship of the human salivary peroxidase system to oral health. J. Oral Pathol. Med. 1984, 13, 573–584.
  2. Nagler, R.M.; Klein, I.; Zarzhevsky, N.; Drigues, N.; Reznick, A.Z. Characterization of the differentiated antioxidant profile of human saliva. Free Radic. Biol. Med. 2002, 32, 268–277.
  3. Uehara, Y.; Kikuchi, K.; Nakamura, T.; Nakama, H.; Agematsu, K.; Kawakami, Y.; Maruchi, N.; Totsuka, K. H2O2 Produced by Viridans Group Streptococci May Contribute to Inhibition of Methicillin-Resistant Staphylococcus aureus Colonization of Oral Cavities in Newborns. Clin. Infect. Dis. 2001, 32, 1408–1413.
  4. Hamon, C.B.; Klebanoff, S.J. A PEROXIDASE-MEDIATED, STREPTOCOCCUS MITIS-DEPENDENT ANTIMICROBIAL SYSTEM IN SALIVA. J. Exp. Med. 1973, 137, 438–450, doi:10.1084/jem.137.2.438.

Reviewer 2 Report

Interesting paper about the stability of flavonoids in saliva.

1.3/1.4: sample preparation - filtration. the possible loss of the analyses due to the filtration, has been evaluated?

1.4: To reduce viscosity and also in order to prevent the sample dilution 1:1, did you evaluated the possibility to introduce a protein precipitation of the saliva?

Sample preparation and LC-MS/MS method: Although an exclusively qualitative method is presented, I recommend introducing a method validation section. there are several international guidelines that it can be followed, on the matter.

For further studies I suggest to introduce also a method for quantitative analysis. 

Author Response

Reviewer 2.

In the beginning, we would like to thank you for the insightful revision of our manuscript. We would like to respond to your comments and reviews. We detailed revisions that have been made, citing the line number and exact change. We hope that you will be satisfied with our comments and corrections.

 1.3/1.4: sample preparation - filtration. the possible loss of the analyses due to the filtration, has been evaluated?

The possible loss of analytes due to the filtration was evaluated. For that purpose standard solution of flavonols were filtered using different filters: PTFE filters (0.45 µm and 0.22 µm), sterile PES filters (0.22 µm) and nylon filters (0.45 µm and 0.22 µm). The evaluation was based on LC-MS analyses of standards of flavonols and comparison the peak. The loss of analyte was not observed in the case of PTFE and sterile PES. Nylon filtered caused the partial loss of analytes, thus the PTFE and PES filters were used in presented analyses.

1.4: To reduce viscosity and also in order to prevent the sample dilution 1:1, did you evaluated the possibility to introduce a protein precipitation of the saliva?

The ratio of saliva and the standard solution was evaluated to obtain optimal flavonols decomposition time which allows detecting intermediates such as depisdes, which are not stable and undergo further decomposition in oxidative conditions. The evaluation of the mentioned ratio was based on a comparison of UV-Vis spectra of quercetin as a representative flavonol in two different ratios 3:1 and 1:1. The 3-fold excess of saliva caused the rapid degradation of quercetin into the dead-end product - protocatechuic acid with no visible band corresponding to depside as an intermediate. This explanation and reference were added in the revised manuscript l. 116-117.

Sample preparation and LC-MS/MS method: Although an exclusively qualitative method is presented, I recommend introducing a method validation section. there are several international guidelines that it can be followed, on the matter.  For further studies I suggest to introduce also a method for quantitative analysis. 

The method validation section was introduced in Apparatus section (2.5) l. 152-167 and due to that, some changes in Table 1 containing the addition of SRM parameters of analysed glycosides were introduced.

Round 2

Reviewer 2 Report

I recommend the publication. 

Thank you